# Evaluation of TiO_2_ Based Photocatalytic Treatment of Odor and Gaseous Emissions from Swine Manure with UV-A and UV-C

**DOI:** 10.3390/ani11051289

**Published:** 2021-04-30

**Authors:** Myeongseong Lee, Jacek A. Koziel, Wyatt Murphy, William S. Jenks, Baitong Chen, Peiyang Li, Chumki Banik

**Affiliations:** 1Department of Agricultural and Biosystems Engineering, Iowa State University, Ames, IA 50011, USA; leefame@iastate.edu (M.L.); wyatt.murphy@jetinc.net (W.M.); baitongc@iastate.edu (B.C.); peiyangl@iastate.edu (P.L.); cbanik@iastate.edu (C.B.); 2Department of Chemistry, Iowa State University, Ames, IA 50011, USA; wsjenks@iastate.edu

**Keywords:** air pollution control, air quality, volatile organic compounds, odor, environmental technology, advanced oxidation, UV

## Abstract

**Simple Summary:**

Poor indoor air quality and gaseous emissions are undesirable side effects of livestock and poultry production. Gaseous emissions of odor, odorous volatile organic compounds (VOCs), ammonia (NH_3_), hydrogen sulfide (H_2_S), and greenhouse gases (GHGs) have detrimental effects on the quality of life in rural communities, the environment, and climate. Proven mitigation technologies are needed to increase the sustainability of animal agriculture. This study’s objective was to evaluate the ultraviolet (UV) light treatment of odor and common air pollutant emissions from stored swine manure on a pilot-scale. To our knowledge, this is the first study of this scope that was needed for scaling up technologies treating gaseous emissions of odor, odorous VOCs, NH_3_, H_2_S, ozone, and GHGs. The study bridged the knowledge gap between lab-scales and simplified treatment of model gases to the treatment of complex gaseous mixtures emitted from swine manure in fast-moving air. The manure emissions were treated in fast-moving air using a mobile lab equipped with UV-A and UV-C lights and photocatalytic surface coating. The percent reduction of targeted gases depended on the UV dose and wavelength. While generally mitigating targeted gases, some UV treatments resulted in CO_2_ and ozone (O_3_). The results proved that the UV technology was sufficiently effective in treating odorous gases, and the mobile lab was ready for farm-scale trials. The UV technology can be considered for the scaled-up treatment of emissions and air quality improvement inside livestock barns.

**Abstract:**

It is essential to mitigate gaseous emissions that result from poultry and livestock production to increase industry sustainability. Odorous volatile organic compounds (VOCs), ammonia (NH_3_), hydrogen sulfide (H_2_S), and greenhouse gases (GHGs) have detrimental effects on the quality of life in rural communities, the environment, and climate. This study’s objective was to evaluate the photocatalytic UV treatment of gaseous emissions of odor, odorous VOCs, NH_3_, and other gases (GHGs, O_3_—sometimes considered as by-products of UV treatment) from stored swine manure on a pilot-scale. The manure emissions were treated in fast-moving air using a mobile lab equipped with UV-A and UV-C lights and TiO_2_-based photocatalyst. Treated gas airflow (0.25–0.76 m^3^∙s^−1^) simulates output from a small ventilation fan in a barn. Through controlling the light intensity and airflow, UV dose was tested for techno-economic analyses. The treatment effectiveness depended on the UV dose and wavelength. Under UV-A (367 nm) photocatalysis, the percent reduction of targeted gases was up to (i) 63% of odor, (ii) 51%, 51%, 53%, 67%, and 32% of acetic acid, propanoic acid, butanoic acid, *p*-cresol, and indole, respectively, (iii) 14% of nitrous oxide (N_2_O), (iv) 100% of O_3_, and 26% generation of CO_2_. Under UV-C (185 + 254 nm) photocatalysis, the percent reductions of target gases were up to (i) 54% and 47% for *p*-cresol and indole, respectively, (ii) 25% of N_2_O, (iii) 71% of CH_4_, and 46% and 139% generation of CO_2_ and O_3_, respectively. The results proved that the UV technology was sufficiently effective in treating odorous gases, and the mobile lab was ready for farm-scale trials. The UV technology can be considered for the scaled-up treatment of emissions and air quality improvement inside livestock barns. Results from this study are needed to inform the experimental design for future on-farm research with UV-A and UV-C.

## 1. Introduction

Poor indoor air quality and gaseous emissions are undesirable side effects of livestock and poultry production. Gaseous emissions of odor, odorous volatile organic compounds (VOCs), ammonia (NH_3_), hydrogen sulfide (H_2_S), and greenhouse gases (GHGs) have detrimental effects on the quality of life in rural communities, the environment, and climate. Proven mitigation technologies are needed to increase the sustainability of animal agriculture. The farm-scale readiness and the effectiveness of technologies for mitigation of gaseous emissions from livestock agriculture are summarized by Maurer et al. [1]. The user-friendly description of technologies and the scientific literature database is provided by the Iowa State University Extension and Outreach website [2].

Photocatalysis with UV (ultraviolet) light has received considerable attention for special applications in indoor air quality. However, the research of UV photocatalysis in livestock agriculture applications is still limited. Costa et al. [3] and Guarino et al. [4] pioneered UV-A photocatalysis in swine weaning and farrowing units reporting mitigation of NH_3_, GHGs, particulate matter (PM), and increased feed conversion efficiency. However, the previous research’s technical design information, such as light dose and photocatalyst coating thickness required for application to actual farms, was not provided. Our team has been motivated by these early examples of farm-scale applications in Europe to conduct lab-to-farm-scale research to scale up and adapt UV photocatalysis to the swine farming systems prevailing in the American swine industry.

Several lessons were learned from the lab-scale to the pilot-scale progression of research. Zhu et al. [5] showed that a TiO_2_ based photocatalytic coating (PureTi, Cincinnati, OH, USA) is sufficient to effectively mitigate odorous VOCs. Research showing the reduction of NH_3_ and odorous VOCs with UV-C followed [6,7,8]. Recently, the application of UV-A photocatalysis for NH_3_, odorous VOCs, ozone, and nitrous oxide (N_2_O) was shown [9,10]. Testing UV-A photocatalysis (a safer bandwidth for direct human and animal exposure) showed a mitigation effect on a pilot-scale in the actual livestock farm environment [9,11].

Thus, earlier tests show practical percent reduction efficiencies for several targeted odorous air pollutants using marketed spray-on coatings for indoor building materials. Still, practical research questions must be addressed before the UV-A (or UV-C) technology can be adopted for farm-scale application to barn interiors to improve air quality inside livestock barns. Additionally, there is an interest in scaling up the UV treatment to mitigate emissions from the barn exhaust air. There is also an interest in UV-C applications to mitigate the risk of airborne pathogens from the ambient air, feed, supplies, personnel threatening farm biosecurity, and using UV to lower the pathogen load inside barns [12]. Thus, this research addresses the gap in knowledge to scale up UV-A and UV-C technology from proven performance mitigating two standard gases (NH_3_, butan-1-ol) at a mobile lab-scale to the treatment of a much more complex mixture of gases released from swine manure. Comprehensive assessment of the mitigation effects for a wide range of gases is needed for scaling up technologies treating gaseous emissions of odor, odorous VOCs, NH_3_, H_2_S, ozone, and GHGs. This research aimed to scale up TiO_2_-based photocatalysis treatment with UV-A and UV-C light to pilot-scale conditions. Specifically, the objective was to evaluate the percent reduction of gaseous emissions and investigate the required UV dose to mitigate the targeted odorous gases generated from swine manure, where the realistic mix of gases and aerosols was treated at fast-moving air and airflows consistent with those on production-scale farms. This study used a mobile UV laboratory designed and commissioned for testing with large (~1 m^3^/s) airflows [13]. This study data obtained under simulated swine conditions with fast-moving airflows, like a real swine farm, is considered helpful in evaluating UV photocatalysis performance. Results from this study are needed to inform the experimental design for future on-farm research with UV-A and UV-C. Box 1 provides definitions of key acronyms used in this paper.

Box 1Definitions of key acronyms used in this paper.**CH_4_**: methane**CO_2_**: carbon dioxide**DMDS**: dimethyl disulfide**DMTS**: dimethyl trisulfide**ECD**: electron capture detector**FID**: flame ionization detector**GC-MS**: gas chromatograph-mass spectrometer**GHGs**: greenhouse gases**H_2_S**: hydrogen sulfide**J:** Joule (unit of energy)**LED**: light-emitting diode**NIST**: National Institute of Standards and Technology**NH_3_**: ammonia**N_2_O**: nitrous oxide**OU_E_**: odor unit**O_3_**: ozone**ppb**: part per billion**ppm:** part per million**SPME**: solid-phase microextraction**TiO_2_**: titanium dioxide**UV-A**: ultraviolet light (315–400 nm range)**UV-C**: ultraviolet light (100–280 nm range)**VOCs**: volatile organic compounds**W**: watt (unit of power)

## 2. Materials and Methods

### 2.1. Materials and Methods

The mobile laboratory (7.2 × 2.4 × 2.4 m) designed and verified in the previous study [13] was used in this study. The mobile laboratory consisted of 12 chambers (7.2 × 0.9 × 2.4 m), and each chamber (0.53 × 0.9 × 2.4 m) was divided into vertical baffles. Chambers #11 and #12 were connected without a vertical baffle. Each chamber was equipped with 11 panels coated with TiO_2_ (nanostructured TiO_2_ anatase at 10 μg/cm^2^ from PureTi, Cincinnati, OH, USA) on all sides. Two fans (I-Fan Type 40, Fancom, Panningen, the Netherlands) were installed on the mobile laboratory to control the airflow inside. The air velocity was measured with the anemometer fan (ATM, Fancom, Panningen, the Netherlands) installed in chamber #10, and the internal airflow can be controlled in real-time using the fan monitoring system (Lumina 20/21, Fancom, Panningen, the Netherlands) by controlling the two fans and the anemometer fan.

### 2.2. Generation of Odorous Gas Emissions from Swine Manure

A plastic drum (55 gal, ~200 L) filled with 35–40 gal of swine manure was used to generate a realistic mixture of odorous gases and aerosols and investigate UV photocatalysis performance (Figure 1 and Figure A1). Compressed air was continuously supplied to the bottom of the manure (Appendix A), and the headspace gas was blended with ambient air. A filtration unit prevented the inflow of flies and dust into the UV mobile lab. Detailed information about the mobile laboratory and filter house has been reported in the previous study [13].

### 2.3. Tested UV Sources

In this study, the mitigation of target gases was investigated using four different light sources (UV-A: 367 nm and UV-C: 254, 222, 185 + 254 nm, Figure A2). Two different low-pressure mercury sources were used, both of which emit strongly at 254 nm, but one additionally contains a small 185 nm component because the bulb is made from special materials that allow transmission of that line. The emission spectrum of low-pressure mercury lamps is well known, and these sources both also contained small emissions at 365 nm and other wavelengths common to all of these bulbs. Nonetheless, we refer to these as 254 nm or (185 + 254) nm light sources. An excimer source emitting at 222 nm was the third source; these three sources constitute variations on wavelengths between 222 and 365 nm. The fourth source was an LED with emission centered at 367 nm, quite near the 365 nm range that mercury lamps commonly were used for, but without many of the disadvantages of a mercury-based lamp. This is considered within the UV-A region.

Each chamber inside the mobile laboratory was equipped with 5 UV-A LED lamps (T8 LED, Eildon Technology, Shenzhen, China). An additional 100 UV-A lamps (effectively adding 20 times the light intensity, Appendix A) were installed on a removable rack in each of the two chambers (#2–#3) to investigate the reduction of targeted gases according to the UV dose (Figure 2). Detailed information on UV-A lamps used in this study was reported previously [13].

All UV-C sources were tested inside chamber #2 while all the UV-A lights in other chambers were turned off. For UV-C (254 nm and 185 + 254 nm, American Ultraviolet Co, Lebanon, IN, USA), four lamps of each different wavelength were installed on the door in one chamber (#2). In the case of the 222 nm excimer UV-C (Ushio America Inc., Cypress, CA, USA), one lamp (Care222 Series) and power supply were installed on the door in chamber #2 (Figure A2). The effects of UV wavelength were measured locally in chamber #2 for all lamp types. The targeted gas concentrations in the untreated gas (control) were measured in the #1 chamber’s sampling port. The treated gas concentration after the UV treatment was measured in the #3 chamber’s sampling port (Figure 1 and Figure 2).

### 2.4. The Light Intensity of Different UV Wavelength Lamps

The light intensity is needed to estimate the UV irradiation (and therefore, the dose when integrated over time). The light intensity was measured by ILT-1700 radiometer (International Light Technologies, Peabody, MA, USA) with wavelength-specific sensors and filters. The UV-C 254 nm, 222 nm, 185 nm, and UV-A 365 nm was measured by the SED240 sensor (w/NS254 filter; 254 ± 5 nm); SED240 sensor (w/NS220 filter, 220 ± 5 nm); SED185 sensor (w/NS185 filter, 185 ± 5 nm); SED033 sensor (w/NS365 filter, 365 ± 5 nm), respectively. The 222 nm sensor only imperfectly excludes light from its intended window, and non-zero artifactual measurements were seen with the two Hg sources. All UV lamps were turned on for 5 min before each measurement or experimental run to ensure stable and consistent UV irradiation. For techno-economic analysis, the electric power consumption was measured using a wattage meter (P3, Lexington, NY, USA). The summary of measured light intensity inside the mobile lab under different UV wavelengths and doses is shown in Appendix A.

### 2.5. Measurement of Odor

Gas samples for odor analyses were collected from the inlet and outlet gas sampling ports inside the UV mobile lab into 10 L Tedlar bags using a Vac-U-Chamber and sampling pump (both from SKC Inc., Eighty-Four, PA, USA). Tedlar bags were precleaned by flushing with clean air three times before use. Gas samples were analyzed for odor using a dynamic triangular forced-choice olfactometer (St. Croix Sensory Inc., Stillwater, MN, USA). Four trained panelists at two repetitions each were used to analyze each sample, presented from low to increasingly lower dilutions to the point of consistent odor detection.

### 2.6. Measurement of Odorous Volatile Organic Compounds

Odorous VOCs, such as sulfur-containing VOCs, volatile fatty acids (VFAs), and phenolic compounds, are significant contributors to livestock odor [14]. VOC analysis was conducted in the same way as described in detail in the previous study [15]. Briefly, VOC samples were collected in 1 L gas sampling glass bulbs. An internal standard (hexane) was used to minimize variability in sampling and sample preparation. A 2 cm DVB/Carboxen/PDMS solid-phase microextraction (SPME) fiber (Supelco, Bellefonte, PA, USA) was used to extract VOCs from the glass bulbs for 50 min, then analyzed with a GC-MS within 12 h of sample collection. The NIST mass spectral library (with at least 80% spectral match) was used to confirm the compounds’ identity. A set of 15 standards for targeted odorous VOC were used (acetic acid, propionic acid, isobutyric acid, butyric acid, isovaleric acid, valeric acid, hexanoic acid, dimethyl disulfide, diethyl disulfide, dimethyl trisulfide, guaiacol, phenol, *p*-cresol, 4-ethyl phenol, indole, and skatole) and calibrated to verify the GC retention time and MS spectral signal.

### 2.7. Measurement of Ozone Concentrations

Ozone is generated during UV-C irradiation of air, and thus, it was a targeted gas. On the other hand, the generated O_3_ can react and mitigate odorous VOCs. In this research, the O_3_ detector was connected to the monitoring system (Series 500 monitor, Aeroqual, New Zealand) and installed at the gas sampling ports when in use. The detector was factory-calibrated (Gas Sensing, Inwood, IA, USA) and certified before use. The detection range was 0–50 ppb.

### 2.8. Measurement of Greenhouse Gas Concentrations

UV treatment of odorous VOCs and NH_3_ can result in the generation of GHGs that should be tracked. Methane (CH_4_), carbon dioxide (CO_2_), and nitrous oxide (N_2_O) were measured. GHGs samples were collected using syringes and 5.9 mL Exetainer vials (Labco Limited, UK) and were analyzed for concentrations on a GC equipped with FID and ECD detectors (SRI Instruments, Torrance, CA, USA). Samples were analyzed on the day of collection. Standard calibrations were constructed daily using 10.3 and 20.5 ppm CH_4_, 1005 and 4010 ppm CO_2_, and 0.101 and 1.01 ppm N_2_O. Pure helium was used to calibrate the baseline of 0 ppm (Air Liquide America, Plumsteadville, PA, USA).

### 2.9. Measurement of Ammonia and Hydrogen Sulfide Concentrations

Ammonia (NH_3_) is a major contributor to air pollution from livestock operations. Hydrogen sulfide (H_2_S) is a toxic air pollutant and a significant contributor to odor. NH_3_ and H_2_S concentrations were measured with a real-time analyzer (OMS-300, Smart Control & Sensing, Daejeon, Korea) calibrated with high precision standard gases (5-point dilution, R^2^ = 0.99). The analyzer was equipped with NH_3_/CR-200 and H_2_S/C-50 electrochemical gas sensors (Membrapor, Wallisellen, Switzerland), NH_3_/CR-200 (0 to 100 ppm), and H_2_S/C-50 (0 to 50 ppm), respectively.

### 2.10. Evaluation of Treatment Effectiveness and Data Analysis

The overall mean percent reduction for each measured targeted gas was estimated using:% R = (C_con_ − C_Treat_)/C_con_ × 100(1)
where C_Con_ and C_Treat_ are the mean measured concentrations in control and treated air, respectively. For odor and odorous VOCs, odor units (OUE∙m^−3^) and MS detector responses (peak area counts, PAC) were used.

Emissions were calculated as a product of measured gas concentrations and the total airflow rate through the UV mobile lab, adjusted for standard conditions and dry air using collected environmental data. The overall mean emission of each measured gas was estimated as:Emission (g·min^−1^) = C × V × (273.15 K × MW)/(273.15 K + T) × (2.24 × 10^4^)(2)
where C = the mean measured target gas concentration in control and treated air (mL/m^3^, OU_E_/m^3^). V = the treated airflow rate (m^3^/min). MW = the molecular weight of the targeted gas (g/mol). T = the temperature in the control and treated air. The 2.24 × 10^4^ is an ideal gas conversion factor for L to moles at 273.15 K [13].

The electric energy consumption during UV treatment was estimated using the measured power consumption by lamps:EEC = P × t_s/_(3600 × 1000)(3)
where EEC = electric energy consumption (kWh). P = measured electric power consumption for the UV lamps turned ‘on’ during treatment (W). t_s_ = treatment time for air irradiated with the UV lamps that were turned ‘on’ inside the mobile lab (s).

The mass of mitigated gas pollutant (M) with UV during given treatment time (t_s_) was estimated by comparing gas emission rate (E) in treatment and control:M = (E_con_ − E_treat_) × t_s/_60(4)
where M = mass of mitigated gas pollutant (g). E_con_ = emission rate at the ‘control’ sampling location. E_treat_ = emission rate at the ‘treatment’ sampling location. 

The electric energy of UV treatment (EE, kWh/g) was estimated as using electric energy consumption (EEC) needed to mitigate a gas pollutant mass (M):EE = EEC/M(5)

Finally, the estimated cost of electric energy (Cost) needed for UV treatment was estimated using the mean cost of rural energy in Iowa (USD 0.13/kWh): Cost = EE × USD 0.13/kWh(6)
where Cost = estimated cost of electric energy needed for UV treatment to mitigate a unit mass of pollutants in the air (USD/g).

UV dose was estimated using measured light intensity (I) at a specific UV wavelength (mW/cm^2^) and treatment time (t_s_). Since the photocatalysis reaction was assumed to be the primary mechanism for the target gas mitigation, the light intensity irradiated on the TiO_2_ surface was used. For lamps emitting light at multiple UV wavelengths, the UV dose was calculated using the light intensity of the primary wavelength suggested by the lamp manufacturer.
UV dose = I × t_s_(7)
where UV Dose = energy of the UV light on the surface of photocatalyst (mJ/cm^2^).

### 2.11. Statistical Analysis

The overall mean percent reduction for each measured targeted gas was estimated using the following: R studio (version 3.6.2; Boston, MA, USA) was used to analyze the target standard gases’ mitigation under UV photocatalysis treatment. The UV dose and treatment time parameters between control concentration and treatment concentration were statistically analyzed using one-way ANOVA. The statistical difference was confirmed by obtaining the *p*-value through the Tukey test. A significant difference was defined for a *p*-value < 0.05 in this study.

## 3. Results

### 3.1. Mitigation of Targeted Gases as a Function of UV-A Dose Controlled by Light Intensity and Airflow Rate

#### 3.1.1. Odor—Effects of UV-A Dose

The UV-A photocatalysis showed a significant percent odor reduction. The UV dose of 2.5 mJ/cm^2^ was required for statistically significant odor reduction (Table 1). As the UV dose was increased, the odor reduction increased up to 63%.

#### 3.1.2. Volatile Organic Compounds—Effects of UV-A Dose

UV-A photocatalysis significantly mitigated selected targeted odorous VOCs while also generating a small subset of other VOCs. This is an important observation as the complex and compound-specific photocatalytic reactions can affect the overall percent reduction of odor. UV-A dose ≥ 2.5 mJ/cm^2^ was required to mitigate phenolic compounds (Table 2), similarly to the findings for odor where the same UV dose resulted in significant mitigation (Table 1). As the UV dose increased up to ~3.9 mJ/cm^2^, the percent reduction of VOCs and the number of mitigated (targeted) VOCs increased. The highest percent reductions were measured for acetic acid (49%), butanoic acid (53%), *p*-cresol (67%), and indole (32%). The highest dose (5.8 mJ/cm^2^) did not improve the mitigation effect, suggesting that there is merit to optimizing the UV dose, especially from the techno-economic standpoint.

#### 3.1.3. Ozone—Effects of UV-A Dose

Compared with a baseline (ambient air) amount of O_3_ detected without UV irradiation, the concentration of O_3_ was effectively mitigated (up to 100%) by UV-A irradiation (Table 3). This observation was consistent with our earlier UV-A research in lab-scale and pilot-scale (poultry farm) conditions [9,10]. Therefore, the treatment of the lowest UV dose (1.3 mJ·cm^−2^) is the most economical condition if O_3_ is the targeted gas. It is also important to mention that the mean O_3_ concentration in the UV mobile lab outlet was 4.7 ppb. This concentration is relatively low and likely of low concern for scaling up to farm environments, where abundant VOCs are present to react with O_3_ and further reduce the risk of its release to the atmosphere outside the barn.

#### 3.1.4. Greenhouse Gases—Effects of UV-A Dose

The N_2_O concentrations were significantly reduced (by 4–14%) with UV-A photocatalysis for 2.5 mJ/cm^2^ or higher doses (Table 4). However, there was no statistically significant change in CH_4_ concentrations (Appendix A), and there was a significant generation of CO_2_ (up to −26%) (Appendix A) that increased with the UV dose.

#### 3.1.5. Ammonia and Hydrogen Sulfide—Effects of UV-A Dose

Significant percent reduction of NH_3_ concentrations was measured only for the maximum UV-A dose (5.8 mJ/cm^2^; Table 5). The treatment efficiency was low (6%), similar to the previous reports [4,9,10,13]. The mean NH_3_ concentration in control was 5.4 ppm. No steady concentration of H_2_S was measured in control (likely due to the limited supply of it in manure). The H_2_S was typically detectable at the start of the experiment, but its concentration in control was rapidly diminishing, preventing reproducible measurements after UV-A treatment. This limitation will be addressed in farm-scale trials, where H_2_S in barn air or barn exhaust is continuously present, the mitigation of H_2_S can be objectively tested.

### 3.2. Comparison of the Mitigation of Targeted Gases as a Function of UV Wavelength

The comparison of UV-A and UV-C photocatalysis treatment was conducted in only one chamber (#2) due to the limited number of available UV-C lamps that are more costly than UV-A. The results are summarized below. Testing conditions were the same for all lamps to enable a fair side-by-side comparison.

#### 3.2.1. Odor—Effects of UV Wavelength

The short UV-C wavelength (185 + 254 nm) resulted in a 44% reduction of overall detected odor. This was a remarkable mitigation effect, considering that the UV dose was the lowest among all tested (Table 6). However, odor reduction was not significant for all treatments (0.09 < *p*-value < 0.94). This was likely due to the variability of control used for just one treatment chamber tested (Table 6). Odor measurements via dilution olfactometry and human panelists are inherently more variable than chemical analyses. This limitation could be addressed by refurbishing the entire UV mobile laboratory with one type of lamp, effectively allowing a more extensive range of doses to be tested (e.g., lower variability reported for UV treatment using an entire mobile lab with 12 chambers facilitating treatment, Table 1).

#### 3.2.2. Volatile Organic Compounds—Effects of UV Wavelength

The phenolic compounds of *p*-cresol and indole were effectively treated with UV-C (185 + 254 nm) with a statistically significant percent reduction at 47 and 54%, respectively (Table 7). *p*-Cresol and indole are often referred to as the ‘signature’ barnyard odors and potent odorants; thus, their mitigation is consistent with the results for overall odor reduction (Table 6). The UV-C (185 + 254 nm) dose was the lowest tested, yet the percent reductions for other targeted VOCs were notable and ranged from 10 to 59%. The 185 + 254 nm light source is essentially identical to the 254 nm light source, save that the ‘glass’ of the lamp itself additionally transmits a small amount of very high energy 185 nm photons. The additional 185 nm irradiation (when part of 185 + 254 nm treatment) results in effective reduction of targeted VOCs. The UV-C (254 and 222 nm) sources also effectively mitigated much targeted VOCs (from 15 to 70%), although there is no significant statistical mitigation. The use of longer-wavelength UV-A (367 nm) and the highest dose resulted in a statistically significant reduction for acetic acid (57%) and butanoic acid (33%).

#### 3.2.3. Ozone—Effects of UV Wavelength

O_3_ was reduced at all wavelengths except for 185 + 254 nm. Specifically, complete mitigation (below detection limits) was measured for 222 and 254 nm treatments. The percent reduction increased from 30 to 97% as the UV dose increased for the 367 nm wavelength. O_3_ increased by ~140% (Table 8) for the 185 + 254 nm treatment. This is due to the direct photolysis of O_2_ in the air, which leads to O_3_ formation.

#### 3.2.4. Greenhouse Gases—Effects of UV Wavelength

Significant mitigation was measured for CH_4_, with the (185 + 254) nm lamps. However, other wavelength lamps did not show statistically significant reduction (Table 9). CO_2_ concentrations increased for all UV wavelengths tested (Appendix A) and were statistically significant for 185 + 254 nm and 367 nm (high UV dose). N_2_O was mitigated at statistically significant levels (from 8 to 25%) for all treatments except for the low 367 nm dose (Table 10). The highest percent reduction for CH_4_ and N_2_O resulted from the 185 + 254 nm treatment.

#### 3.2.5. Ammonia and Hydrogen Sulfide—Effects of UV Wavelength

There was no statistically significant reduction for all UV wavelengths tested, and the percent reduction ranged from 0.3 to 2.1% (Table 11). The average concentration of the control group was 3.1 ppm (Table 11). H_2_S concentrations in control were not stable enough to warrant reporting the effect.

## 4. Discussion

### 4.1. Summary of the UV-A Photocatalysis—Comparison with Previous Research

UV photocatalysis can be considered a potential technology to reduce odorous gases and improve air quality. This research provides mitigation data for a more extensive set of odorants and air pollutants compared with the state-of-the-art. UV-A photocatalysis reduced several of the targeted odorous gases (Table 12) with statistical significance. The reproducibility of mitigation with UV-A photocatalysis warrants further scaling up into larger volumetric flowrates common for farm applications.

This research provided data that can be considered for early assessment and extrapolating the techno-economic analysis of the UV-A treatment to practical scales (Table 13).

Caution needs to be exercised when extrapolating pilot-scale data. However, several major recommendations can be made. The UV-A treatment does not appear to be effective for farm-scale mitigation of NH_3_, considering that the mitigation effect was rather small (Table 5 and Table 11). Thus, effective reduction of kg/day quantities of NH_3_ from typical swine farms with UV-A appears to be too costly.

On the other hand, mitigation of several targeted air pollutants is worth considering. For example, mitigation of N_2_O (the most potent GHG, Table 4 and Table 10) might be further exploited for farm income generation that uses subsidies and programs focused on reducing GHGs emissions and mitigating climate change. Direct emissions of O_3_ from farms have not been a concern, as opposed to the secondary pollutant generation of O_3_ as a by-product of emitted VOCs and their atmospheric chemistry. Thus, the incentivization and credit taking for the at-source mitigation of O_3_ might be considered (Table 3 and Table 8). Finally, the significant reduction of odor and odorous VOCs is encouraging at this scale (Table 1, Table 2, Table 6 and Table 7). Of course, planned farm-scale trials can provide a more realistic techno-economic assessment of UV-A cost. Farm-scale trials with the UV-A photocatalysis installed inside barns to mitigate indoor air quality and the pathogen load are warranted.

### 4.2. Summary of the UV-C Photocatalysis

The effects of UV wavelength were only tested in one chamber inside the UV mobile lab due to increasing UV-C dose limitation. Thus, the comparison is somewhat limited (Table 14). Therefore, the results show the UV-C’s future potential that still needs to be tested on a larger scale. The UV-C can efficiently reduce odorous VOCs with a lower dose (compared to UV-A). One caveat to UV-C use is risks associated with direct skin and eye tissue exposure and O_3_ generation. 

### 4.3. Evaluation of UV Photocatalysis Based on TiO_2_ in the Livestock Environment

We summarized the percent mitigation of targeted gases in the previous studies and this mobile lab research series to show the UV photocatalysis performance (Lee et al., 2021; Table 14) [16]. The mitigation of selected target gases via photocatalysis with UV-A and UV-C in livestock-relevant environmental conditions can be considered as an effective method to mitigate the odorous gases.

The TiO_2_ based photocatalysis with UV-A yields significant reductions of NH_3_ (~31%), H_2_S (~40%), CH_4_ (~27%), N_2_O (~14%), O_3_ (~100%), Odorous VOCs (~100%), and odor (~63%) [16]. In the case of CO_2_, generation has been reported after UV-A photocatalysis in previous studies. CO_2_ is the oxidative endpoint for photocatalytic oxidation of virtually all carbon-containing compounds, and thus its mitigation would not derive from its chemical removal. The percent reduction for the targeted gas showed a difference depending on the coating thickness and UV dose.

The UV-C photocatalyst showed a higher mitigation effect at a lower dose than UV-A photocatalysis. In particular, it showed a significant reduction in H_2_S (~100%), CH_4_ (~40%) and VOCs (~100%) even after irradiation for a relatively short time (1 s) [6,17,18]. Additionally, it is encouraging that it can effectively reduce H_2_S, which is harmful to farms, among the compounds generated in swine barns. In previous research results, it was reported that UV photocatalysis showed high efficiency compared to other mitigation technologies in economic analysis (estimated average electricity cost of UV treatment per pig was USD 0.15–0.23) [6,18].

## 5. Conclusions

This study evaluated the photocatalytic UV-A and UV-C treatment of gaseous emissions of odor, odorous VOCs, NH_3_, and other gases (GHGs, O_3_) from stored swine manure on a pilot-scale. To our knowledge, this is the first study of this scope that was needed for scaling up technologies treating gaseous emissions of odor, odorous VOCs, NH_3_, H_2_S, ozone, and GHGs. The study bridged the knowledge gap between lab-scales and simplified treatment of model gases to the treatment of complex gaseous mixtures emitted from swine manure in fast-moving air. The results showed that the proposed UV technology is ready for the next stage of testing and mitigation of emissions from swine farms. The treatment effectiveness depended on the UV dose and wavelength. Specific findings are summarized below. 

Under UV-A (367 nm) photocatalysis, the percent reduction of targeted gases was up to:63% of odor,51%, 51%, 53%, 67%, and 32% of acetic acid, propanoic acid, butanoic acid, *p*-cresol, and indole, respectively,14% of nitrous oxide (N_2_O),100% of O_3_, 6% of NH_3_, and26% generation of CO_2_.

Under UV-C (185 + 254 nm) photocatalysis, the percent reduction of target gases was up to:54% and 47% for *p*-cresol and indole, respectively,25% of N_2_O,71% of CH_4,_ and46% and 139% generation of CO_2_ and O_3_, respectively.

UV-C (222 nm) photocatalysis showed a reduction of 8% for N_2_O, 100% for O_3_. Lastly, UV-C (254 nm) photocatalysis showed a reduction of 14% for N_2_O, 100% for O_3_. The UV-A photocatalysis (367 nm) was not economical to reduce NH_3_; while it appeared to be economical and effective in mitigating odor and VOC. The 2.5 mJ/cm^2^ dose is required to significantly reduce odor. UV-C photocatalysis (185 + 254 nm) was shown to be more efficient than UV-A photocatalysis by significantly reducing several target gases with a low dose, but additional research is needed because there was a limit to the dose control of UV-C in this study. The results proved that the UV technology was sufficiently effective in treating odorous gases in a simulated swine emissions environment, and the mobile lab was ready for farm-scale trials. The UV technology can be considered for the scaled-up treatment of emissions and air quality improvement inside livestock barns.

## Figures and Tables

**Figure 1 animals-11-01289-f001:**
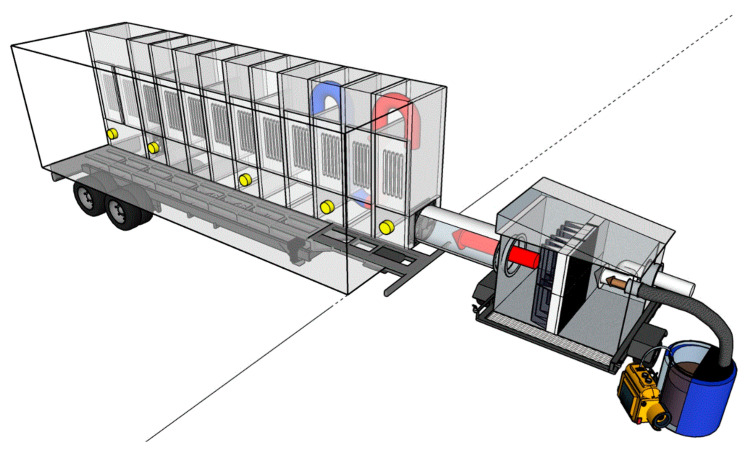
Schematic of mobile laboratory for UV treatment of gaseous emissions. UV dose is controlled by either adjusting treatment time (by controlled airflow rate) or adjusting irradiation (by turning lamps on/off). Brown arrow: untreated gas from the manure drum (blue barrel); white arrow: ambient air for diluting the untreated gas; red arrow: inlet air to UV treatment with reduced particle matter load (due to the filtration unit; pictured on the right); blue arrow: UV-treated air. Yellow: gas sampling ports.

**Figure 2 animals-11-01289-f002:**
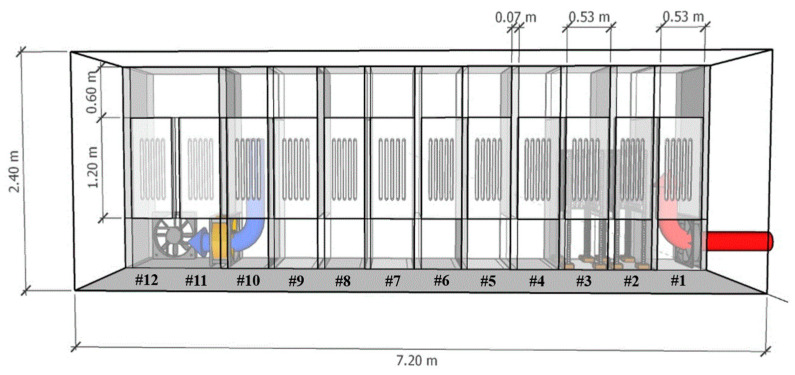
Detailed schematic (side view) of UV treatment inside a flow-through mobile laboratory with UV lamps. The untreated airflow is irradiated while passing through a series of chambers (#1–#12) equipped with TiO_2_ photocatalytic surfaces and 5 UV-A lamps per chamber. The two chambers (#2 and #3) were equipped with additional 100 portable lamp holders to increase light intensity. Treated air moves in a serpentine pattern from the inlet (right, red) to the outlet (left, blue). UV dose is controlled by either adjusting treatment time (by controlled airflow rate) or adjusting irradiation (by turning lamps on/off).

**Table 1 animals-11-01289-t001:** Mitigation of odor with the different UV-A doses (1.3, 2.0, 2.5, 3.9, and 5.8 mJ/cm^2^) irradiating gaseous emissions from swine manure. Bold signifies statistical significance.

UV-A Dose (mJ/cm^2^)	Light Intensity (mW/cm^2^)	Treatment Time (s)	Control (OU_E_/m^3^)	Treatment (OU_E_/m^3^)	% Reduction (*p*-Value)
UV dose control with light intensity					
1.3	0.14	9.5	378 ± 13	229 ± 75	39.4 (0.12)
2.5	0.26	9.5	352 ± 8.0	239 ± 24	**32.2 (0.04)**
3.9	0.41	9.5	653 ± 32	277 ± 22	**57.5 (0.01)**
UV dose control with treatment time					
1.3	0.41	3.2	198 ± 59	234 ± 60	−18.6 (0.61)
2.0	0.41	4.8	212 ± 31	206 ± 75	2.9 (0.93)
UV dose control with light intensity and treatment time					
5.8 *	0.41 and 0.04	9.5 and 47.6	653 ± 9.2	243 ± 64	**62.7 (<0.01)**

Note: * Irradiation with 5 UV-A lamps per each chamber (#1–#12, a total of 60 lamps turned on) with additional 100 portable UV-A lamps turned on in chambers #2 and #3, 160 lamps turned on total. Inlet and outlet air temperature = 19 ± 2 °C and 22 ± 5 °C.

**Table 2 animals-11-01289-t002:** Mitigation of odorous VOCs with the different doses (1.3, 2.0, 2.5, 3.9, and 5.8 mJ/cm^2^) irradiating gaseous emissions from swine manure. Bold signifies statistical significance.

Targeted VOCs	Percent Reduction (*p*-Value)					
	UV-A Dose (mJ/cm^2^) (UV Light Intensity, mW/cm^2^ and Treatment Time, s)					
	1.3 (0.14 and 9.5)	1.3 (0.41 and 3.2)	2.0 (0.41 and 4.8)	2.5 (0.26 and 9.5)	3.9 (0.41 and 9.5)	5.8 ^*^ (0.41 + 0.04 & 9.5 + 47.6)
DMDS	29.3 (0.36)	54.3 (0.12)	41.9 (0.46)	59.2 (0.07)	−21.8 (0.51)	8.3 (0.92)
DMTS	−5.8 (0.96)	−5.6 (0.84)	−0.9 (0.99)	11.5 (0.35)	42.9 (0.07)	49.4 (0.22)
Acetic acid	−1.3 (0.97)	−10.8 (0.85)	23.9 (0.69)	−4.0 (0.91)	**48.6 (0.04)**	**50.5 (0.04)**
Propanoic acid	12.7 (0.36)	21.1 (0.07)	72.9 (0.30)	**50.7 (0.01)**	76.7 (0.40)	66.8 (0.45)
Isopentanoic acid	29.4 (0.39)	49.4 (0.27)	54.2 (0.23)	24.2 (0.70)	41.9 (0.10)	37.7 (0.27)
Butanoic acid	1.5 (0.95)	**36.3 (0.04)**	**35.8 (0.01)**	**44.8 (<0.01)**	**52.6 (0.04)**	**47.9 (0.04)**
Phenol	39.1 (0.15)	34.0 (0.07)	63.1 (0.26)	−7.9 (0.86)	−28.1 (0.13)	−32.1 (0.44)
*p*-Cresol	−3.2 (0.96)	36.1 (0.05)	−15.9 (0.84)	**41.3 (0.03)**	**66.5 (0.03)**	58.6 (0.05)
Indole	0.2 (0.99)	4.0 (0.89)	23.3 (0.61)	21.5 (0.58)	**32.3 (0.02)**	**20.0 (0.03)**
Skatole	−9.2 (0.89)	17.6 (0.06)	6.4 (0.93)	6.4 (0.93)	70.0 (0.37)	64.6 (0.45)

Note: DMDS = dimethyl disulfide, DMTS = dimethyl trisulfide; values in table report percent reduction (*p*-values). * Irradiation with 5 UV-A lamps per each chamber (#1–#12, a total of 60 lamps turned on) with additional 100 portable UV-A lamps turned on in chambers #2 and #3, 160 lamps turned on total. Inlet and outlet air temperature = 19 ± 2 °C and 22 ± 5 °C.

**Table 3 animals-11-01289-t003:** Mitigation of O_3_ with the different UV doses (1.3, 2.0, 2.5, 3.9, and 5.8 mJ/cm^2^) irradiating gaseous emissions from swine manure. Bold signifies statistical significance.

UV-A Dose (mJ/cm^2^)	Light Intensity (mW/cm^2^)	Treatment Time (s)	Control (ppb)	Treatment (ppb)	% Reduction (*p*-Value)
UV dose control with UV light intensity					
1.3	0.14	9.5	2.9 ± 0.3	0.0 ± 0.0	**100 (<0.01)**
2.5	0.26	9.5	3.8 ± 0.1	0.0 ± 0.0	**100 (<0.01)**
3.9	0.41	9.5	9.5 ± 1.2	0.0 ± 0.0	**100 (<0.01)**
UV dose control with treatment time					
1.3	0.41	3.2	5.8 ± 2.0	0.0 ± 0.0	**100 (<0.01)**
2.0	0.41	4.8	2.3 ± 0.2	0.0 ± 0.0	**100 (<0.01)**
UV dose control with UV light intensity and treatment time					
5.8 ^*^	0.41 and 0.04	9.5 and 47.6	3.0 ± 0.2	0.0 ± 0.0	**100 (<0.01)**

Note: * Irradiation with 5 UV-A lamps per each chamber (#1–#12, a total of 60 lamps turned on) with additional 100 portable UV-A lamps in chambers #2 and #3 turned on, 160 lamps total (installed and portable). Inlet and outlet air temperature = 19 ± 2 °C and 22 ± 5 °C.

**Table 4 animals-11-01289-t004:** Mitigation of N_2_O with the different UV doses (1.3, 2.0, 2.5, 3.9, and 5.8 mJ/cm^2^) irradiating gaseous emissions from swine manure. Bold signifies statistical significance.

UV-A Dose (mJ/cm^2^)	Light Intensity (mW/cm^2^)	Treatment Time (s)	Control (ppm)	Treatment (ppm)	% Reduction (*p*-Value)
UV dose control with UV light intensity					
1.3	0.14	9.5	0.4 ± 0.0	0.4 ± 0.0	3.5 (0.22)
2.5	0.26	9.5	0.4 ± 0.0	0.3 ± 0.0	**9.0 (<0.01)**
3.9	0.41	9.5	0.3 ± 0.0	0.3 ± 0.0	**4.3 (0.02)**
UV dose control with treatment time					
1.3	0.41	3.2	0.3 ± 0.0	0.3 ± 0.0	0.4 (0.85)
2.0	0.41	4.8	0.3 ± 0.0	0.2 ± 0.0	17.1 (0.09)
UV dose control with UV light intensity and treatment time					
5.8 ^*^	0.41 and 0.04	9.5 and 47.6	0.3 ± 0.0	0.2 ± 0.0	**14.2 (0.03)**

Note: * Irradiation with 5 UV-A lamps per each chamber (#1–#12, a total of 60 lamps turned on) with additional 100 portable UV-A lamps in chambers #2 and #3 turned on, 160 lamps total (installed and portable). Inlet and outlet air temperature = 19 ± 2 °C and 22 ± 5 °C.

**Table 5 animals-11-01289-t005:** Mitigation of NH_3_ with the different UV doses (1.3, 2.0, 2.5, 3.9, and 5.8 mJ/cm^2^) irradiating gaseous emissions from swine manure. Bold signifies statistical significance.

UV-A Dose (mJ/cm^2^)	Light Intensity (mW/cm^2^)	Treatment Time (s)	Control (ppm)	Treatment (ppm)	% Reduction (*p*-Value)
UV dose control with UV light intensity					
1.3	0.14	9.5	4.6 ± 0.1	4.5 ±0.1	1.0 (0.33)
2.5	0.26	9.5	5.8 ± 0.1	5.7 ±0.1	1.3 (0.21)
3.9	0.41	9.5	5.5 ± 0.2	5.3 ±0.2	2.1 (0.38)
UV dose control with treatment time					
1.3	0.41	3.2	4.3 ± 0.4	4.2 ± 0.4	4.3 (0.41)
2.0	0.41	4.8	6.3 ± 0.1	6.2 ± 0.1	2.9 (0.93)
UV dose control with UV light intensity and treatment time					
5.8 ^*^	0.41 and 0.04	9.5 and 47.6	6.0 ± 0.3	5.6 ± 0.3	**6.1 (0.04)**

Note: * Irradiation with 5 UV-A lamps per each chamber (#1–#12, a total of 60 lamps turned on) with additional 100 portable UV-A lamps in chambers #2 and #3 turned on, 160 lamps total (installed and portable). Inlet and outlet air temperature = 19 ± 2 °C and 22 ± 5 °C.

**Table 6 animals-11-01289-t006:** Mitigation of odor with different UV wavelengths irradiating gaseous emissions inside the #2 chamber.

UV Wavelengths (nm)	UV Dose (µJ/cm^2^)	Light Intensity (µW/cm^2^)	Control (OU_E_/m^3^)	Treatment (OU_E_/m^3^)	% Reduction (*p*-Value)
185 + 254	0.03	0.01	325 ± 78	182 ± 73	43.9 (0.09)
222	2.83	0.59		262 ± 22	19.5 (0.30)
254	1.78	0.37		290 ± 55	10.6 (0.19)
367	192	40		332 ± 58	−2.2 (0.94)
	1968	410		270 ± 38	17.0 (0.11)

Note: Treatment time = 4.8 s (airflow = 0.25 m^3^/s), inlet and outlet air temp. = 16 ± 1 °C and 19 ± 2 °C.

**Table 7 animals-11-01289-t007:** Mitigation of odorous VOCs with different UV wavelength irradiating gaseous emissions inside the #2 chamber. Bold signifies statistical significance.

Targeted VOCs	Percent Reduction (*p*-Value)				
	UV-C Dose, µJ/cm^2^ (Light Intensity, µW/cm^2^)			UV-A Dose (Light Intensity, µW/cm^2^)	
	185 + 254 nm 0.03 (0.01)	222 nm 2.83 (0.59)	254 nm 1.78 (0.37)	367 nm 192 (40)	367 nm 1968 (410)
DMDS	59.3 (0.14)	59.8 (0.27)	14.8 (0.70)	−14.7 (0.87)	−4.1 (0.93)
DMTS	56.4 (0.15)	67.1 (0.12)	21.7 (0.50)	−6.2 (0.63)	5.8 (0.86)
Acetic acid	10.0 (0.68)	−12.2 (0.72)	−12.1 (0.70)	0.4 (0.99)	**57.2 (0.04)**
Propanoic acid	13.3 (0.83)	37.4 (0.47)	23.1 (0.64)	−32.6 (0.54)	36.2 (0.49)
Isopentanoic acid	24.9 (0.72)	60.5 (0.39)	70.0 (0.33)	−13.6 (0.85)	18.1 (0.80)
Butanoic acid	10.3 (0.66)	27.8 (0.08)	−15.5 (0.79)	21.8 (0.39)	**33.4 (0.03)**
Phenol	43.4 (0.08)	32.4 (0.26)	7.8 (0.79)	23.2 (0.53)	24.7 (0.52)
*p*-Cresol	**47.1 (0.04)**	46.9 (0.09)	29.2 (0.32)	8.9 (0.75)	46.8 (0.05)
Indole	**54.2 (0.01)**	19.2 (0.48)	16.9 (0.51)	−14.5 (0.64)	46.6 (0.17)
Skatole	35.1 (0.35)	55.8 (0.14)	64.6 (0.11)	3.7 (0.83)	56.5 (0.09)

Note: DMDS = dimethyl disulfide, DMTS = dimethyl trisulfide. Treatment time = 4.8 s (airflow = 0.25 m^3^/s), inlet and outlet air temperature = 16 ± 1 °C and 19 ± 2 °C.

**Table 8 animals-11-01289-t008:** Mitigation of O_3_ concentration with different UV wavelength irradiating gaseous emissions inside the #2 chamber. Bold signifies statistical significance.

UV Wavelength (nm)	UV Dose (µJ/cm^2^)	Light Intensity (µW/cm^2^)	Control (ppb)	UV Treatment (ppb)	% Reduction (*p*-Value)
185 + 254	0.03	0.01	14.6 ± 4.2	34.8 ± 5.7	**−139 (<0.01)**
222	2.83	0.59	18.6 ± 2.2	0.0 ± 0.0	**100 (<0.01)**
254	1.78	0.37	16.8 ± 2.9	0.0 ± 0.0	**100 (<0.01)**
367	192	40	10.9 ± 2.1	7.7 ± 2.4	**30 (0.02)**
	1968	410	6.4 ± 1.6	0.2 ± 0.7	**97 (<0.01)**

Note: Treatment time = 4.8 s (airflow = 0.25 m^3^/s), inlet and outlet air temp. = 16 ± 1 °C and 19 ± 2 °C.

**Table 9 animals-11-01289-t009:** Mitigation of CH_4_ concentration with different UV wavelength irradiating gaseous emissions inside the #2 chamber. Bold signifies statistical significance.

UV Wavelength (nm)	UV Dose (µJ/cm^2^)	Light Intensity (µW/cm^2^)	Control (ppm)	Treatment (ppm)	% Reduction (*p*-Value)
185 + 254	0.03	0.01	3.7 ± 0.9	1.1 ± 0.0	**70.9 (0.04)**
222	2.83	0.59		1.7 ± 0.0	55.7 (0.06)
254	1.78	0.37		1.4 ± 0.2	63.6 (0.06)
367	192	40		3.9 ± 0.5	−3.7 (0.77)
	1968	410		3.2 ± 1.3	15.9 (0.59)

Note: Treatment time = 4.8 s (airflow = 0.25 m^3^/s), inlet and outlet air temp. = 16 ± 1 °C and 19 ± 2 °C.

**Table 10 animals-11-01289-t010:** Mitigation of N_2_O concentration with different UV wavelength irradiating gaseous emissions inside the #2 chamber. Bold signifies statistical significance.

UV Wavelength (nm)	UV Dose (µJ/cm^2^)	Light Intensity (µW/cm^2^)	Control (ppm)	Treatment (ppm)	% Reduction (*p*-Value)
185 + 254	0.03	0.01	0.24 ± 0.01	0.18 ± 0.00	**25.4 (<0.01)**
222	2.83	0.59		0.22 ± 0.00	**8.1 (0.01)**
254	1.78	0.37		0.21 ± 0.01	**13.6 (0.01)**
367	192	40		0.23 ± 0.01	5.9 (0.17)
	1968	410		0.21 ± 0.00	**13.5 (0.02)**

Note: Treatment time = 4.8 s (airflow = 0.25 m^3^/s), inlet and outlet air temp. = 16 ± 1 °C and 19 ± 2 °C.

**Table 11 animals-11-01289-t011:** Mitigation of NH_3_ concentration with different UV wavelength irradiating gaseous emissions inside the #2 chamber.

UV Wavelength (nm)	UV Dose (µJ/cm^2^)	Light Intensity (µW/cm^2^)	Control (ppm)	Treatment (ppm)	% Reduction (*p*-Value)
185 + 254	0.03	0.01	3.3 ± 0.1	3.3 ± 0.0	0.5 (0.55)
222	2.83	0.59	3.2 ± 0.1	3.2 ± 0.1	1.3 (0.35)
254	1.78	0.37	3.1 ± 0.1	3.0 ± 0.1	1.4 (0.22)
367	192	40	2.5 ± 0.1	2.5 ± 0.1	0.3 (0.71)
	1968	410	2.6 ± 0.1	2.6 ± 0.1	2.1 (0.16)

Note: Treatment time = 4.8 s (airflow = 0.25 m^3^/s), inlet and outlet air temp. = 16 ± 1 °C and 19 ± 2 °C.

**Table 12 animals-11-01289-t012:** Summary and comparison of the % reduction of targeted gases with UV-A photocatalysis. Bold signifies statistical significance.

Reference	Targeted Gas	UV Dose (mJ/cm^2^)	Target Gas Concentration (ppm, O_3_ = ppb, Odor = OU_E_/m^3^)		% Reduction
			Control	UV Treatment	
[13] (pilot-scale treating standard gases)	NH_3_	3.9	67.4 ± 0.36	61.1 ± 0.30	**9**
		5.8	68.9 ± 0.68	61.1 ± 0.70	**11**
	Butan-1-ol	2.5	0.66 ± 0.02	0.53 ± 0.06	**19**
		3.9	0.65 ± 0.03	0.43 ± 0.04	**34**
		5.8	0.69 ± 0.02	0.41 ± 0.07	**41**
This study (pilot-scale with swine manure)	NH_3_	5.8	5.98 ± 0.28	5.62 ± 0.34	**6**
	N_2_O	3.9	0.29 ± 0.01	0.28 ± 0.00	**4**
	N_2_O	5.8	0.29 ± 0.00	0.25 ± 0.01	**14**
	O_3_	1.3	0.34 ± 0.03	0	**100**
	O_3_	5.8	0.31 ± 0.02	0	**100**
	VOCs	2.5	N/A	N/A	**PA(51), BA(45), *p*-Cresol(41)**
		3.9	N/A	N/A	**AA(49), BA(53), *p*-Cresol (67), Indole(32)**
		5.8	N/A	N/A	**AA(51), BA(48), Indole(20)**
	Odor	2.5	352 ± 7.98	239 ± 24.4	**32**
		3.9	653 ± 32.1	277 ± 21.7	**58**
		5.8	653 ± 9.25	243 ± 64.4	**63**

Note: DMDS = dimethyl disulfide, acetic acid = AA, propanoic acid = PA, isobutyric acid = IA, and butanoic acid = BA, N/A = not available.

**Table 13 animals-11-01289-t013:** Techno-economic analysis of mitigating target gases with UV-A photocatalysis.

Reference	Targeted Gas	UV Dose (mJ/cm^2^)	Target Gas Emission (*E*, mg/min, Odor, OU_E_/min)		Cost ^1^ (USD/kg for NH_3_, USD/g for butan-1-ol, N_2_O, O_3_ USD/ton of OU_E_ for Odor)
			Control	UV Treatment	
[13] (pilot-scale treating standard gases)	NH_3_	3.9	746	676	53.4
	NH_3_	5.8	763	676	62.5
	Butan-1-ol	2.5	31.5	25.3	442
		3.9	30.9	20.3	352
		5.8	32.9	19.4	403
This study (pilot-scale treating emissions from swine manure)	NH_3_	5.8	64.2	60.2	1260
	N_2_O	3.9	8.14	7.79	10.6
	N_2_O	5.8	8.06	6.92	4.72
	O_3_	1.3	0.01	0.00	18.9
	O_3_	5.8	0.01	0.00	60.0
	Odor	3.9	9200	3910	0.71
		5.8	9200	3430	0.94

Note: ^1^ electric energy needed for UV treatment to mitigate a unit mass of pollutants in the air (USD/g).

**Table 14 animals-11-01289-t014:** Summary of the % reduction of targeted gases with UV-C photocatalysis. Bold signifies statistical significance.

UV Wavelength, nm (UV dose, µJ/cm^2^)	Targeted Gas	Target Gas Concentration (ppm; O_3_ = ppb)		% Reduction
		b	UV Treatment	
185 + 254 (0.03)	VOCs	N/A	N/A	***p*-Cresol (47); Indole (54)**
	CH_4_	3.7 ± 0.9	1.1 ± 0.0	**71**
	N_2_O	0.2 ± 0.0	0.2 ± 0.0	**25**
222 (2.80)	N_2_O	0.2 ± 0.0	0.2 ± 0.0	**8**
	O_3_	18.6 ± 2.2	0.0 ± 0.0	**100**
254 (1.76)	N_2_O	0.2 ± 0.0	0.2 ± 0.0	**14**
	O_3_	16.8 ± 2.9	0.0 ± 0.0	**100**

Note: N/A = not available.

## Data Availability

The original contributions presented in the study are included in the article/Appendix A; further inquiries can be directed to the corresponding author.

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
