# Peer review of "Evaluation of TiO2 Based Photocatalytic Treatment of Odor and Gaseous Emissions from Swine Manure with UV-A and UV-C"

_animals, 2021, doi:10.3390/ani11051289_

Round 1

Reviewer 1 Report

This work entitled "The Evaluation of TiO2 Based Photocatalytic Treatment of Odor and Gaseous Emissions From Swine Manure With UV-A and UV-C" examines scalability of TiO2-based photocatalysis treatment with UV-A and UV-C light to pilot-scale conditions. The paper can be acceptable for publication after major revise.

  • The introduction is well written. However, I suggest that authors include a separate paragraph at the end of the introduction explaining the significance of the work as well as objectives. This could be also added to the abstract section.
  • Please provide a table of abbreviations as there is a lot of terms abbreviated throughout the text.
  • Conclusion section appears to be just a detailed summary of results/observations. All conclusions must be convincing statements on what was found to be novel, impactful based on strong support of the data/results/discussion. Conclusions must improve, Original and unique contribution explicit in and quantified the Conclusions (this should not be another Abstract).
  • Results and discussion part is basically more results. I would suggest that the authors compare their results with recent publications in the field and enhance the number of references. This would add value to this work.
  • To what extent this approach is feasible to implement in other types of barns (cattle barns, chicken barns) and if so, does it need huge modification? Perhaps it could be addressed in the conclusion section.

Reviewer 2 Report

The manuscript evaluated the photocatalytic UV-A/UV-C effect on odor, VOCs, ammonia, GHG, ozone ecc from swine manure.The application of UV-A, UV-C with different wavelength, UV dose and light intensity was tested.

The topic is extremely interesting, for the importance to reduce pollutants originated and emitted from confinements, storage sites and fertilized soil in livestock farming.

The manuscript was well designed and clearly presented. 

An important result is that N2O, with the highest global warming potential among the considered GHG, was mitigated significantly (from 8 to 25%) almost for all treatments (not for 367 nm). Good the economic analysis for the treatments application to livestock buildings.

Observations:

At L 59 correct the citations : "...pioneered UV-A photocatalysis in swine weaning and farrowing units reporting mitigation of NH3, GHGs, particulate matter (PM)…."

Line 94 Please correct the dimensions of the chambers (0.53 m, as reported in the scheme of Figure 2)

Reviewer 3 Report

The work of dr. Myeongseong Lee et al. aimed to evaluate the photocatalytic UV treatment of gaseous emissions of odor, odorous volatile organic compounds (VOCs), ammonia (NH3), and other gases  (greenhouse gases (GHGs), O3 – sometimes considered as by-products of UV treatment) from stored swine manure on  a pilot-scale because they have detrimental effects on the quality of life in rural communities, the environment, and climate.

The manure emissions were treated in fast-moving air using a mobile lab equipped with UV-A and UV-C lights and TiO2-based photocatalyst. Treated gas airflow (0.25 to 0.76 m3∙s−1)  simulates output from a small ventilation fan in a barn. Through controlling the light intensity and airflow, UV dose was tested also from the  techno-economic point of view.

They have shown that the treatment effectiveness depended on the UV dose and wavelength. The proposed technology UV technology was sufficiently effective in treating odorous gases, and the mobile lab is ready for farm-scale trials. The UV technology can be considered for the scaled-up treatment of emissions and air quality improvement inside livestock barns.

Author Response

This manuscript is a resubmission of an earlier submission. The following is a list of the peer review reports and author responses from that submission.

Round 1

